# microRNA Biomarkers in Paediatric Infection Diagnostics—Bridging the Gap Between Evidence and Clinical Application: A Scoping Review

**DOI:** 10.3390/ncrna11050071

**Published:** 2025-09-24

**Authors:** Oenone Rodgers, Anna De Beer, Thomas Waterfield

**Affiliations:** 1Wellcome-Wolfson Institute for Experimental Medicine at Queen’s University Belfast, Belfast BT9 7BL, UK; t.waterfield@qub.ac.uk; 2Barts and the London School of Medicine and Dentistry, Queen Mary University of London, London E1 2AD, UK; a.debeer@smd24.qmul.ac.uk

**Keywords:** microRNA, paediatric, novel biomarkers, acute infection, bacterial, viral, febrile

## Abstract

Background: Distinguishing between bacterial and viral infections in children remains a significant challenge for clinicians. Traditional biomarkers have limited utility, often leading to antibiotic overprescription due to clinician uncertainty. With rising antimicrobial resistance, novel biomarkers are needed to improve diagnosis. This scoping review examines current host miRNA biomarkers for acute bacterial and viral infections in children (0–18), focusing on study methods, diagnostic metrics, and research gaps to support clinical translation. Results: Of the 1147 articles identified, 36 studies were included. Notably, 72.2% of the studies originated from Asia, and the distribution across the paediatric age groups was relatively even. A total of 17 miRNAs were validated in at least two independent studies. Three miRNAs, hsa-miR-182-5p, hsa-miR-363-3p, and hsa-miR-206, were consistently associated with bacterial infection in children. Meanwhile, nine miRNAs were associated with viral infections: hsa-miR-155, hsa-miR-29a-3p, hsa-miR-155-5p, hsa-miR-150-5p, hsa-miR-140-3p, hsa-miR-142-3p, hsa-miR-149-3p, hsa-miR-210-3p, and hsa-miR-34a-5p. Across the 12 studies reporting diagnostic accuracy metrics, miRNA biomarkers exhibited a sensitivity ranging from 70% to 100%, and a specificity ranging from 72% to 100%. The area under the curve across the studies demonstrated a range from 0.62 to 0.99. Conclusions: This scoping review highlights the potential of miRNA targets for diagnosing paediatric infections when studied rigorously. However, clinical translation is limited by poor adherence to STARD guidelines, lack of robust diagnostic metrics, and study heterogeneity. Many studies were set up with a case–control design, a design that, while highlighting differences, is more likely to identify non-specific biomarkers rather than those that are useful for novel clinical diagnostics.

## 1. Introduction

Febrile children present a significant challenge across global healthcare systems. Fever is the most common reason for children to present to Emergency Departments (ED) [1]. In the paediatric European population, antibiotic overprescription is an issue, with prescribing rates approaching 64%, with only approximately 13% benefiting from these prescriptions [1,2]. The global overuse of antibiotics in children results in increased healthcare costs and worsening antibiotic resistance [3]. These issues do not just occur in paediatrics, with global estimates suggesting that antibiotic prescribing has increased by 16.3% between 2016 and 2023 [4]. The most pronounced increases were in upper-middle and low-middle-income countries [4]. It is projected that by 2030 global antibiotic consumption will increase by a further 52.3% above 2023’s value [4]. In 2021 there were 193,000 global paediatric deaths attributed to antimicrobial resistant organisms [5], with a notable increase in antimicrobial resistant paediatric sepsis, where 31.3% of cases were associated with drug-resistant bacterial infections [5]. This surge in antibiotic resistance coupled with the absence of novel antimicrobials is a substantial threat to public health [6].

Clinicians overprescribe antibiotics primarily due to uncertainty. Differentiating between bacterial and viral infections is challenging due to the significant overlap in clinical presentations. The gold standard for diagnosing sterile site bacterial infections is via culturing. However, this process has a long wait time for results and is limited by issues with contamination and difficult-to-culture pathogens [7]. Molecular testing for pathogens has also been shown to be poorly specific, with positive viral testing results often representing colonisation rather than active infection [8]. Therefore, clinicians often rely on biomarkers to ensure appropriate treatment and antimicrobial stewardship efforts for febrile children. The most studied biomarkers are C-reactive protein (CRP) and procalcitonin (PCT) [9]. Clinically, biomarkers can be assessed in numerous biofluid samples, particularly from blood, cerebrospinal fluid, and urine [9]. The most used blood-based biomarker is CRP; however, it is not bacterial infection-specific and substantial elevations occur with other inflammatory or infectious causes [9]. Furthermore, CRP has been found not to improve the selection of a low-risk group of children at ED presentation [10]. Screening with PCT is recommended by three major global clinical practice guidelines [11,12,13]. However, a meta-analysis of the performance of PCT to that of CRP for serious bacterial infections in young infants suggests that they hold similar diagnostic accuracy metrics [14]. PCT is not a biomarker specific to bacterial infection, and its lack of superiority over CRP has led to NICE guidelines not recommending its routine use in the UK [15]. Therefore, the lack of a reliable biomarker leads to uncertainty and over prescribing of antibiotics ‘just in case’.

The focus for paediatric infection diagnostics revolves around circulating biomarkers, as they are relatively non-invasive and easily integrated into clinical workflows. Any new diagnostic tool must therefore offer a like-for-like replacement or be combined with current testing to be viable in practice. The search for the ideal biomarker for paediatric infection diagnostics has largely underperformed [16]. Previous efforts primarily targeted single protein biomarkers, but researchers are now exploring biomarker panels that include both proteins and other biomolecules. The growing application of omics approaches has led to the identification of many potential biomarker candidates across a range of clinical disciplines; however, their integration into routine clinical practice has been slow [17,18]. Significant hurdles have impacted the success of novel diagnostics for clinical use. These include large discrepancies in the diagnostic criteria used for clinical research and the consequential variations in biomarker discovery results and reliability. Furthermore, when promising candidates have been identified, these markers are rarely validated in large external cohorts [19,20]. Even when validation studies have been conducted, a key barrier to clinical implementation is the lack of protocol consistency in validation studies. This prevents meaningful cross-study comparisons, impedes biomarker prioritisation, and ultimately stagnates clinical biomarker outputs. A further complication arises from the high demands placed on clinical biomarkers. To be considered for implementation, candidates must demonstrate substantial improvements over existing tests in heterogeneous populations, while also offering a favourable cost–benefit profile. In the context of infection diagnostics, this means outperforming established markers such as CRP and PCT, which are inexpensive, readily available, and often considered ‘good enough’ for continued clinical use. This is highlighted by the routine use of PCT in some healthcare systems, particularly in the United States. However, PCT is not routinely implemented in the UK due to a lack of cost–benefit ratio supporting widespread use.

Former work has identified host-RNA signatures in paediatric infectious disease [21] that show promise for alternative routes to infection differentiation. Among these alternative routes are microRNA (miRNA), short non-coding RNAs that regulate post-transcriptional gene expression, which may offer novel diagnostic insight. Compared to classical protein biomarkers, miRNAs present several advantages for paediatric infection diagnostics. They demonstrate high sensitivity and specificity, undergo rapid changes in abundance in response to disease, and are readily accessible through minimally invasive sampling of biofluids such as blood, saliva, or urine [22]. Furthermore, miRNAs demonstrate excellent physicochemical stability within biofluid samples, particularly in whole blood, serum, and plasma [23,24], which makes miRNAs particularly suitable for clinical diagnostics. Of particular interest is the timing of miRNA abundance changes, which may occur more rapidly than the protein biomarkers in current clinical use. This characteristic could make miRNAs especially valuable for early diagnosis in paediatric hospital triage, where timely clinical decisions are critical. By enabling faster identification of infectious causes, miRNA-based diagnostics have the potential to reduce delays commonly associated with conventional methods such as blood cultures or screening via CRP or PCT which tend to rise late in the disease course. However, miRNA as targets for bacterial and viral differentiation have not been fully established.

The literature on paediatric infection diagnostics utilising miRNA targets is highly varied and relatively scarce. Given the early stage of research in this area and the broad scope of the research question, a scoping review is an appropriate methodological approach to the topic. This scoping review aims to summarise the published human miRNA signatures associated with acute bacterial and viral infections in children. This review also aims to examine the methodologies and diagnostic metrics used across studies to identify research gaps and support the advancement of clinically translatable applications.

## 2. Results

### 2.1. Search Selection and Study Characteristics

A total of 1147 articles were identified: 302 articles from the Ovid Medline All database, 1125 articles from the Web of Science Core Collection, and a singular article from the grey literature search. A total of 138 duplicate records were removed, leaving 1290 articles to be screened. A total of 1209 articles were removed by title and abstract; this culminated in 81 articles for full-text review. Following the full-text review, 45 articles were excluded, resulting in 36 articles to be included within the study. Reasons for exclusion during the full-text review are presented in the PRISMA-ScR flow diagram (Figure 1).

The included articles were published between 2012 and 2025, all of which were cohort studies. Studies were predominantly conducted in China (20/36; 55.6%) [25,26,27,28,29,30,31,32,33,34,35,36,37,38,39,40,41,42], the United States (8/36; 22.2%) [43,44,45,46,47,48,49,50], Japan (2/36; 5.5%) [51,52], Czech Republic (1/36; 2.8%) [53], Italy (1/36; 2.8%) [54], Thailand (1/36; 2.8%) [55], Taiwan (1/36; 2.8%) [56], Norway (1/36; 2.8%) [57], India (1/36; 2.8%) [58], Egypt (1/36; 2.8%) [59], and Singapore (1/36; 2.8%) [60]. Within these, 35 (97%) articles recruited patients who were hospitalised [25,26,27,28,29,30,31,32,33,34,35,36,37,38,39,40,41,42,43,44,45,46,47,49,50,51,52,53,54,55,56,57,58,59,60] and 1 (2.8%) article recruited non-hospitalised children [48].

### 2.2. Characteristics of Populations

The characteristics of the studies and populations are shown in Table 1. The eligible studies involved a total of 3885 subjects, ranging from 6 to 575 subjects per study. The percentages of female subjects per study ranged from 14% to 67%. Within the 36 included studies, children aged from newborn to 13 years were investigated. For the purposes of this study, neonates were classified as 0 to 28 days old, infants were classified as 29 to 364 days old, toddlers were classified as 1 to 5 years old, and children were classified as 5 to 13 years old. Within these categories there were 9 neonatal studies (9/36; 25%) [26,28,34,47,48,52,53,58,59], 7 infant studies (7/36; 19.4%) [25,36,40,44,46,54,57], 9 toddler studies (9/36; 25%) [33,38,39,45,49,50,51,56,60], and 11 studies on children (11/36; 30.6%) [27,29,30,31,32,35,37,41,42,43,55]. The majority of the studies compared infection to healthy controls (24/36; 66.6%) [28,29,30,32,33,34,35,36,37,38,39,40,45,49,52,53,54,57,58,60], while fewer investigated infection to another disease state (9/36; 25%) [26,27,31,43,46,47,50,51,55], and an additional three studies investigated the severity of disease (3/36; 8.3%) [25,44,56].

A total of 14 studies included bacterial infections (14/36; 38.8%) [26,27,28,29,31,32,34,37,40,43,47,53,58,59], and 23 studies included viral infections (23/36; 63.8%) [25,30,33,35,36,38,39,41,42,43,44,45,46,48,49,50,51,52,54,55,56,57,60], of which 1 study included bacterial and viral infections (2.8%) [43]. The bacterial infections included within the studies were as follows: sepsis (173/449; 38.5%), *Mycoplasma pneumoniae* (129/449; 28.7%), *Bordetella pertussis* (66/449; 14.7%), methicillin-resistant *Staphylococcus aureus* (30/449; 6.7%), *Staphylococcus aureus* (19/449; 4.2%), *Escherichia coli* (7/449; 1.6%), *Acinetobacter baumannii* (5/449; 1.1%), *Klebsiella* species (4/449; 0.9%), *Enterococcus* (3/449; 0.7%), *Neisseria meningitidis* (2/449; 0.4%), Group A Streptococcus (2/449; 0.4%), Group B Streptococcus (2/449; 0.4%), pneumococcal meningitis (2/449; 0.4%), Salmonella serotype Typhi (1/449; 0.2%), and *Listeria monocytogenes* (1/449; 0.2%).

The viral infections included within the studies were as follows: Respiratory syncytial virus (1026/1738; 59%), Rhinovirus (218/1738; 12.5%), Respiratory syncytial virus and Rhinovirus coinfection (69/1738; 4%), Enterovirus (62/1738; 3.6%), Dengue virus (60/1738; 3.5%), Hand, foot, and mouth disease (59/1738; 3.4%), Adenovirus (53/1738; 3%), Herpesviridae species (45/1738; 2.6%), Varicella-zoster virus (39/1738; 2.2%), Epidemic B virus (22/1738; 1.3%), Human metapneumovirus (22/1738; 1.3%), Epstein–Barr virus (16/1738; 0.9%), Congenital cytomegalovirus (13/1738; 0.7%), Parotid virus (11/1738; 0.6%), Parainfluenza (11/1738; 0.6%), Influenza (11/1738; 0.6%), and Haemophilus influenza (1/1738; 0.1%).

### 2.3. Reference Standards

The reference standards for diagnosis and categorisation of the patients were as follows: six studies relied on clinical guidelines and clinical diagnosis alone (6/36; 16.7%) [27,31,46,48,56,58]. Laboratory-confirmed infection was used as the reference standard for 13 studies (13/36; 36.1%) [29,32,37,40,43,45,49,50,51,53,54,57,60], and 17 studies used combined laboratory-confirmed infection with clinical diagnosis (17/36; 47.2%) [25,26,28,30,33,34,35,36,38,39,44,47,52,55]. The laboratory-confirmed infection methods were as follows: PCR (13/30; 43.3%) [25,29,33,36,41,42,44,45,49,50,51,52,59,60], antibodies (4/30; 13.3%) [30,32,35,38], blood culture (5/30; 16.6%) [26,34,47,53], PCR with antigen/antibody/culture (4/30; 13.3%) [37,39,40,57], antigen alone (1/30; 3.3%) [54], and laboratory tests completed but specifics not mentioned (3/30; 10%) [28,43,55].

Of the six the studies relying only on clinical guidelines and clinical diagnosis alone, two studies [27,31] relied on clinical guidelines (2/6; 33.3%). These guidelines were the diagnostic criteria for pneumonia in the 2016 clinical practice guidelines of the American Society of Infectious Diseases and the American Thoracic Society [61], and the diagnostic criteria for sepsis was based on the Expert Consensus on the Diagnosis and Treatment of Septic Shock (Infectious Shock) in Children (2015 Edition) [62]. The remaining four studies [46,48,56,58] relied on clinical diagnosis for bronchiolitis, diagnosis of pneumonia, prescription for albuterol, prescription of oral steroids, clinical symptoms of enterovirus infection, or clinically diagnosed neonatal sepsis.

### 2.4. Study Methodologies

Within the 36 included studies, RT-qPCR was the most common methodology (27/36; 75%). The majority of the studies utilised relative RT-qPCR (26/27; 96.3%), with the remaining study utilising absolute RT-qPCR. This was followed by microarray (10/36; 27.7%), then next-generation sequencing (8/36; 22.2%), protein detection (5/36; 13.8%), and viral PCR testing (1/36; 2.8%). Table 2 summarises the methodologies of the included studies.

**Table 2 ncrna-11-00071-t002:** Included studies’ methodologies, RNA extraction kit preferences, NGS kit preferences, sample type preferences, and normalisation methods.

Author [Citation]	Year	Study Methods	RNA Extraction Kit	NGS Kit	Sample Type (Volume)	Normalisation
Jone, et al. [43]	2020	Microarray	Own Methods	N/A	Serum (3 mL)	hsa-miR-320
Zhang, et al. [25]	2020	NGS + RT-qPCR	Trizol Reagent	NEBNext Ultra RNA Library Prep Kit for Illumina	Whole blood	U6
Zhang, et al. [26]	2021	RT-qPCR	Trizol Reagent	N/A	Whole blood	U6
Tian, et al. [27]	2021	RT-qPCR	Trizol Reagent	N/A	BALF	U6
Mao, et al. [28]	2021	RT-qPCR	Trizol Reagent	N/A	Serum	U6
Liu, et al. [30]	2024	RT-qPCR	miRNeasy Serum Kit	N/A	Serum	U6
Kyo, et al. [44]	2024	NGS	Direct-zol RNA Miniprep Kit	NEXTFLEX^®^ small RNAseq v3 kit with Unique Dual Indexes	Nasal + Nasopharyngeal	N/A
Yin, et al. [29]	2021	Microarray + RT-qPCR	Trizol Reagent	N/A	Plasma	Actb
Arroyo, et al. [45]	2020	Viral PCR, + miRNA RT-qPCR	MirVana miRNA isolation kit	N/A	Nasal	cel-mir-39
Janec, et al. [53]	2024	Luminex^®^ and NGS	miRNeasy Serum/Plasma Kit	NEXTFlex small RNA-Seq v3	Plasma (100 µL)	N/A
Zhu, et al. [46]	2023	NGS	Direct-zol RNA Miniprep Kit	NEXTFLEX^®^ small RNA-seq v3 kit with Unique Dual Indexes	Nasal + Nasopharyngeal	N/A
Savino, et al. [54]	2023	RT-qPCR	Promega Simply RNA Blood Kit	N/A	Throat swab (1 mL) + Heparinised blood (200 µL)	RNU43
Ernst, et al. [47]	2021	Microarray	miRNeasy Serum/Plasma Advanced Kit	N/A	Cord plasma + Cord tissue	Global normalisation
Huang, et al. [31]	2021	RT-qPCR + ELSIA	miRNeasy Serum/Plasma Kit	N/A	Plasma (3–5 mL)	U6
Beheshti, et al. [48]	2023	NGS + ELLA	miRNeasy Kit	Illumina TruSeq Small RNA Prep protocol	Saliva	N/A
Jia, et al. [32]	2023	RT-qPCR + ELISA	Trizol Reagent	N/A	Plasma	U6
Sriprapun, et al. [55]	2023	RT-qPCR	miRNeasy mini kit	N/A	Serum	hsa-miR-16-5p
Torii, et al. [51]	2022	NGS	miRCURY Exosome Cell/Urine/CSF kit + miRCURY Exosome Serum/Plasma kit	NEBNext Multiplex Small RNA Library Prep Set for Illumina	CSF (31–1418 mL) + Serum (120 mL)	hsa-miR-204-5p
Qi, et al. [33]	2014	Microarray + RT-qPCR	mirVana PARIS kits	N/A	Serum (400 µL)	cel-miR-238
Chen, et al. [34]	2014	Microarray + RT-qPCR	Trizol Reagent	N/A	Leukocytes	used but does not mention
Wang, et al. [56]	2016	Microarray + RT-qPCR	mirVana PARIS kit	N/A	Serum (100 µL)	Global Normalisation + U6
Gao, et al. [35]	2015	RT-qPCR	miRNeasy serum/plasma kit + miRNeasy mini kit	N/A	Plasma (200 µL) + PBMCs	U6 + cel-miR-39
Liu, et al. [36]	2015	RT-qPCR	Trizol Reagent	N/A	PBMCs	GAPDH + U6
Inchley, et al. [57]	2015	Microarray + RT-qPCR	miRNeasy mini kit	N/A	Nasal	RNU24
Dhas, et al. [58]	2018	RT-qPCR*	miRNeasy Serum/Plasma kit	N/A	Plasma (500 µL)	cel-miR-39
Kawano, et al. [52]	2016	RT-qPCR	mirVana PARIS Kit	N/A	Plasma	hsa-miR-16
Gutierrez, et al. [49]	2016	Microarray	SeraMir kit	N/A	Nasal	Global normalisation
Min, et al. [60]	2018	RT-qPCR	Biofluid extraction kit (Exiqon, Inc., Woburn, MA, USA)	N/A	Throat swab + Saliva	hsa-miR-23a-3p
Hasegawa, et al. [50]	2018	Microarray + NGA	Norgen RNA/DNA Purification Kit		Nasal	Global Normalisation
Li, et al. [37]	2019	RT-qPCR	mirVanaTM miRNA Isolation Kit	N/A	PMBCs	U6
Huang, et al. [38]	2019	NGS + RT-qPCR	Trizol Reagent	NEBNext^®^ Multiplex Small RNA Library Prep Set for Illumina^®^	Serum exosomes	N/A
Huang, et al. [39]	2018	NGS + RT-qPCR	RiboPure™ Blood RNA Isolation Kit	Own Method	Whole blood	U6
Cui, et al. [40]	2012	Microarray + RT-qPCR	mirVana PARIS kit	N/A	Serum (400 μL)	cel-miR-238
Gao, et al. [42]	2024	RT-qPCR	Trizol Reagent	N/A	Serum	U6
Hamdy, et al. [59]	2024	RT-qPCR	miRNeasy Serum/Plasma Kit	N/A	Serum	hsa-miR-16-5p
Qi, et al. [41]	2025	RT-qPCR	Trizol Reagent	N/A	Serum	U6

### 2.5. RNA Extraction Methods

The RNA extraction methodology most commonly used was Trizol reagent, with no reference to specific kits (11/36; 30.6%), followed by the miRNeasy Serum/Plasma Kit (7/36; 19.4%), mirVana PARIS Kit (6/36; 16.7%), miRNeasy Mini Kit (3/36; 8.3%), Direct-zol RNA Miniprep Kit (2/36; 5.6%), miRNeasy Serum/Plasma Advanced Kit (1/36; 2.8%), miRCURY Exosome Cell/Urine/CSF kit (1/36; 2.8%), miRCURY Exosome Serum/Plasma Kit (1/36; 2.8%), Promega simply RNA Blood Kit (1/36; 2.8%), System Biosciences SeraMir Kit (1/36; 2.8%), Exiqon biofluid extraction Kit (1/36; 2.8%), Norgen RNA/DNA Purification Kit (1/36; 2.8%), RiboPure™ Blood RNA Isolation Kit (1/36; 2.8%), and one study using their own methodology (1/36; 2.8%).

### 2.6. Next-Generation Sequencing Methods

Studies investigating samples via next-generation sequencing used a variety of methodologies, with PerkinElmer NEXTFLEX^®^ small RNAseq v3 kit being the most commonly used (3/8; 37.5%), followed by NEBNext Multiplex Small RNA Library Prep Set for Illumina (2/8; 25%), NEBNext Ultra RNA Library Prep Kit for Illumina (1/8; 12.5%), Illumina TruSeq Small RNA Prep protocol (1/8; 12.5%), and one study using its own methodology (1/8; 12.5%).

### 2.7. Sample Type and Volume

The most common sample type recruited for the studies was serum (10/36; 27.8%), followed by nasal secretions, throat swabs, and saliva (9/36; 25%), plasma (9/36; 25%), whole blood (6/36; 16.7%), bronchoalveolar lavage (1/36; 2.8%), and cerebrospinal fluid (1/36; 2.8%). Sample volumes were only reported for 11/36 studies: for plasma samples the median volume was 200 µL (range from 100 µL to 5000 µL), and for serum samples the median volume was 800 µL (range from 100 µL to 3000 µL). There was also one study reporting a throat swab sample volume of 1 mL [54], and another reporting 1 mL of cerebrospinal fluid [51].

### 2.8. Normalisation miRNA

A total of 31 studies referenced a methodology for normalisation. The majority of the studies referenced a specific reference molecule (26/31; 83.9%), whereas studies using global mean normalisation methods were less common (4/31; 12.9%). Six of the studies reported to use human endogenous miRNA (6/31; 19.4%). Endogenous normalisers included the following: hsa-miR-320, hsa-miR-204-5p, hsa-miR-23a-3p, and hsa-miR-16(-5p), and were used three times. A total of 17 studies (17/31; 54.8%) reported using non-miRNA references, of which U6 accounted for 14 of the studies (14/31; 45.2%). Five studies reported using spiked-in RNA (5/31; 16.1%) including cel-miR-39 and cel-miR-238. One study reported using normalisation methodology but was unspecific in their description (1/31; 3.2%).

### 2.9. Included Current Blood-Based Biomarkers

Of the 36 included studies, 13 (36.1%) reported to have incorporated current clinical biomarkers of infection, including CRP and PCT. CRP was mentioned in all 13 of the studies (100%), whereas PCT was only referred to in 3 studies (23.1%).

### 2.10. Novel miRNA Biomarkers Identified

Discovered within the 36 included studies, there were 164 human miRNAs identified with paediatric infection. There were 16 miRNAs that were identified more than once across the studies, shown in Table 3: hsa-miR-155 (6/36; 16.7%), hsa-miR-29 (5/36; 13.9%), hsa-miR-206 (4/36; 11.1%), hsa-miR-142-3p (3/36; 8.3%), hsa-miR-182-5p (3/36; 8.3%), hsa-miR-155-5p (2/36; 5.5%), hsa-miR-101-3p (2/36; 5.5%), hsa-miR-140-3p (2/36; 5.5%), hsa-miR-142-5p (2/36; 5.5%), hsa-miR-142-3p (2/36; 5.5%), hsa-miR-150-5p (2/36; 5.5%), hsa-miR-183-5p (2/36; 5.5%), hsa-miR-210-3p (2/36; 5.5%), hsa-miR-34 (2/36; 5.5%), hsa-miR-363-3p (2/36; 5.5%), and hsa-miR-486-3p (2/36; 5.5%). These duplications account for 47 miRNAs from the total of 164 identified, and the remaining 117 miRNAs were only identified once across the studies.

### 2.11. Bacterial Infection-Associated miRNA

Table 4 summarises the miRNA associations with acute bacterial infections. From the 14 studies that included bacterial samples, 64 miRNAs were identified, with 3 of the miRNAs being validated across two separate studies. hsa-miR-182-5p (2/14; 14.3%), hsa-miR-363-3p (2/14; 14.3%), and hsa-miR-206 (2/14; 14.3%) were all identified with neonatal sepsis [47,53,59].

Predominantly, the studies investigating bacterial samples compared infection to healthy controls (9/14; 64.3%), whereas fewer comparisons were made between infection and other disease (5/14; 35.7%). Most of the studies related to paediatric or neonatal sepsis (8/14; 57.1%), followed by studies on pneumonia and respiratory infection (5/14; 35.7%), and one study comparing Kawasaki Disease to febrile controls with bacterial infection (1/14; 7.1%).

**Table 4 ncrna-11-00071-t004:** miRNA associations with bacterial infection.

Author [Citation]	Biomarkers			Clinical Question	Study Methods	Ages	Sample Size	CRP/PCT	Reference Standard
Jone et al. [43]	miR-210-3p miR-184 miR-19a-3p			Kawasaki and Febrile Non-Kawasaki	Microarray	Febrile = 3.5–13 years Kawasaki = 1.8–5.4 years	n = 113	CRP	Lab confirmed
Zhang et al. [26]	miR-101-3p PCT			Neonatal Sepsis vs. Respiratory Infection/Pneumonia	RT-qPCR	Neonatal sepsis = 11.88 days Control = 11.21 days	n = 148	CRP + PCT	Lab confirmed + clinical
Tian et al. [27]	miR-155 IL-9			MRSA Pneumonia vs. Bronchial Foreign Bodies	RT-qPCR	<18 years	n = 40		Guidelines
Mao et al. [28]	miR-455-5p			Neonatal Sepsis vs. Healthy	RT-qPCR	Neonatal sepsis = 10.21 days Control = 11.41 days	n = 178	CRP + PCT	Lab confirmed + clinical
Yin et al. [29]	miR-1323 IL-6			Mycoplasma Pneumoniae Pneumonia vs. Healthy	Microarray + RT-qPCR	MPP =7.1 years Control = 6.9	n = 57	CRP	Lab confirmed
Janec et al. [53]	miR-136-3p miR-142-5p miR-148b-3p	miR-182-5p miR-183-5p miR-223-3p	miR-363-3p miR-629-5p miR-3613-5p	Neonatal Sepsis vs. Healthy	Luminex^®^ sRNA-sequencing	1 day	n = 20		Lab confirmed
Ernst et al. [47]	miR-211-5p miR-223-5p miR-331-5p miR-181d-5p miR-146b-5p	miR-142-3p miR-193a-5p miR-532-5p miR-363-3p	miR-15a-5p miR-202-3p miR-206 miR-150-5p	EOS vs. Presumed Sepsis vs. Healthy	Microarray	1 day	n = 41		Lab confirmed + Clinical
Huang et al. [31]	miR-497 FABP3	GPBB cTnI		Sepsis vs. Sepsis and Myocardial Injury vs. Healthy	RT-qPCR + ELSIA	median age 6.9 years	n = 132	CRP + PCT	Clinical Guidelines
Jia et al. [32]	miR-492 TNF-α	IL6 IL-18		Mycoplasma Pneumoniae Pneumonia vs. Healthy	RT-qPCR + ELSIA	median age 6.4 years	n = 86	CRP	Lab confirmed
Chen et al. [34]	miR-208a miR-489 miR-618 miR-644 miR-491-3p miR-450a miR-509-3-5p miR-720	miR-886-3p miR-486-3p miR-1184 miR-375 miR-637 miR-548a-3p miR-185 miR-761	miR-620 miR-122 miR-549 miR-135a miR-611 miR-551b miR-1299	Neonatal Sepsis vs. Healthy	Microarray + RT-qPCR	median age 12.5 days	n = 48		Lab confirmed + clinical
Dhas et al. [58]	miR-223 miR-132			EOS vs. Healthy	RT-qPCR	<3 days old	n = 50	CRP	Clinical only
Li et al. [37]	miR-29			Mycoplasma Pneumoniae Pneumonia vs. Healthy	RT-qPCR	median age 6.4 years	n = 78	CRP	Lab confirmed
Cui et al. [40]	miR-202 miR-342-5p miR-576-5p	miR-206 miR-487b		B. Pertussis vs. Healthy	Microarray + RT-qPCR	median age 1 year	n = 134		Lab confirmed
Hamdy et al. [59]	miR-182-5p miR-590-3p			Neonatal Sepsis vs. Healthy	PCR	<21 days	n = 110	CRP	Lab confirmed

### 2.12. Viral Infection-Associated miRNA

Table 5 summarises the miRNA associations with acute viral infections. From the 23 studies that included viral samples, 100 miRNAs were identified. Ten of these miRNAs of were validated by at least two separate studies: hsa-miR-155 (5/23; 21.7%), hsa-miR-29a-3p (4/23; 17.4%), hsa-miR-155-5p (2/23; 8.7%), hsa-miR-150-5p (2/23; 8.7%), hsa-miR-140-3p (2/23; 8.7%), hsa-miR-142-3p (2/23; 8.7%), hsa-miR-149-3p (2/23; 8.7%), hsa-miR-210-3p (2/23; 8.7%), and hsa-miR-34a-5p (2/23; 8.7%).

The majority of the studies investigating viral samples compared infection to healthy controls (14/23; 60.9%), whereas fewer studies compared infection to infection, or included severity of the same infection (8/23; 37.8%). One study (1/23; 4.3%) was also a prospective cohort study with no comparator group. A large proportion of the viral studies focused on respiratory infections (16/23; 69.6%), whereas the remaining studies were systemic viral infections or not specifically grouped (7/23; 30.4%).

**Table 5 ncrna-11-00071-t005:** miRNA associations with viral infection.

Author [Citation]	Year	Biomarkers			Clinical Question	Study Methods	Ages	Sample Size	CRP/PCT	Reference Standard
Jone et al. [43]	2020	miR-210-3p miR−184 miR-19a-3p.			Kawasaki and Febrile Non-Kawasaki	Microarray	Febrile = 3.5–13 years Kawasaki = 1.8–5.4 years	n = 113	CRP	Lab confirmed
Zhang et al. [25]	2020	miR-1271-5p miR-10a-3p miR-125b-5p	miR-100-5p miR-30b-3p		Mild or Severe RSV-Associated Pneumonia	NGS + RT-qPCR	Median age 6.22 months	n = 46	CRP	Lab confirmed + clinical
Liu et al. [30]	2024	miR-142-3p			Viral Encephalitis vs. Healthy	RT-qPCR	Median age 5.74 years	n = 196		Lab confirmed + clinical
Kyo et al. [44]	2024	miR-991-5p miR-125b-2-3p miR-99b-3p			Severity of RSV Bronchiolitis	NGS	Median age 3 months	n = 493		Lab confirmed + clinical
Arroyo et al. [45]	2020	miR-155			Viral Respiratory Infections vs. Healthy	RT-qPCR + protein detection	Median age 1.2 years	n = 151		Lab confirmed
Zhu et al. [46]	2023	miR-29 miR-22–3p			Severe Bronchiolitis: No Asthma vs. With Asthma	NGS	Median age 3 months	n = 575		Clinical only
Savino et al. [54]	2023	miR-155			RSV vs. Healthy	RT-qPCR	Median age 93 days	n = 66		Lab confirmed
Beheshti et al. [48]	2023	miR-140-3p miR-22-5p miR-29 miR-34c-5p	miR-125a-5p miR-27b-3p miR-203a-3p miR-155-5p		URTIs in the first 12 months after birth	NGS + protein detection	<1 year	n = 146		Clinical only
Sriprapun et al. [55]	2023	miR-126-3p			Dengue vs. Acute Febrile Illness	RT-qPCR	Median age 10.5 years	n = 90		Lab confirmed + clinical
Torii et al. [51]	2022	miR-381-3p miR-155 miR-148a miR-664a-3p	let-7b-5p let-7i-5p miR-483-5p miR-151a	miR-499a-5p miR-140-3p miR-206-3p	Human Herpesvirus 6 (HHV-6): Acute Encephalopathy vs. Febrile Seizures	NGS	Median age 15 months	n = 15		Lab confirmed
Qi et al. [33]	2014	miR-197 miR-363 miR-629	miR-132 miR-122		Varicella vs. Healthy	Microarray + RT-qPCR	Median age 1 year	n = 102		Lab confirmed + clinical
Wang et al. [56]	2016	miR-494 miR-29 miR-551a	miR-606 miR-876-5p miR-30c-5p	miR-221-3p miR-150-5p	EV71 (Severe + Mild) vs. Healthy	Microarray + RT-qPCR	Median age 2.6 years	n = 12		Clinical only
Gao et al. [35]	2015	miR-155-5p miR-34a-5p miR-18b-5p	miR-181a-5p miR-142-5p miR-134-5p	miR-18b-5p miR-34a-5p miR-196a-5p	EBV vs. Healthy	RT-qPCR	Median age 5.4 years	n = 30		Lab confirmed + clinical
Liu et al. [36]	2015	miR-26			RSV vs. Healthy	RT-qPCR	<1 year	n = 40		Lab confirmed + clinical
Inchley et al. [57]	2015	miR-155 miR-31 miR-203a	miR-16 let-7d		RSV vs. Healthy	Microarray + RT-qPCR	<1 year	n = 61		Lab confirmed
Kawano et al. [52]	2016	miR-183-5p miR-210-3p			CMV vs. Healthy	RT-qPCR	Median age 25 days	n = 23		Lab confirmed + clinical
Gutierrez et al. [49]	2016	miR-155 miR-21			RV vs. non-RV	Microarray	Median age 1.5 years	n = 20		Lab confirmed
Min et al. [60]	2018	miR-221			HEV vs. Healthy	PCR	<5 years	n = 59		Lab confirmed
Hasegawa et al. [50]	2019	miR-149-3p miR-197-3p miR-197-5p miR-296-3p	miR-149-3p miR-504-3p miR-155-5p		Sole Rhinovirus Infection vs. Sole RSV Infection	Microarray + protein detection	<6 months	n = 32		Lab confirmed
Huang et al. [37]	2019	miR-103b-5p miR-450a-5p	miR-98-5p miR-103a-3p		HAdV vs. Healthy	NGS + RT-qPCR	Median age 2.5 years	n = 59		Lab confirmed + clinical
Huang et al. [39]	2018	miR-381-3p miR-486-5p miR-409-3p miR-486-3p miR-127-3p miR-182-5p miR-99b-5p	miR-379-5p miR-370-3p let-7e-5p miR-493-5p miR-494-3p miR-101-3p miR-142-3p	miR-150-5p miR-29a-3p miR-186-5p miR-27a-3p miR-342-3p miR-29b-3p	HAdV vs. Healthy	NGS + RT-qPCR	Median age 2.2 years	n = 6	CRP	Lab confirmed + clinical
Gao et al. [42]	2024	miR-425-3p NEAT1			Viral Induced Myocarditis vs. Healthy	RT-qPCR	3–8 years	n = 108	CRP	Lab confirmed + clinical
Qi et al. [41]	2025	miR-200b			Pneumonia vs. Healthy	RT-qPCR	2–10 years	n = 218	CRP	Lab confirmed + clinical

Table 6 summarises the infection groupings per infection sub-type. From the 100 miRNAs associated with viral infection, 68% can be grouped into an infection type. There were 14 miRNAs (14/68; 20.6%) associated with respiratory syncytial virus, 24 (24/68; 35.3%) associated with human adenovirus infection, 8 (8/68; 11.8%) associated with rhinovirus infection, and 22 (22/68; 32.4%) associated with systemic viral infection. In contrast, of the 64 miRNAs associated with bacterial infection, 59 (90.6%) can be grouped into an infection type. A total of 50 miRNAs (50/59; 84.7%) were associated with sepsis, and 9 (9/59; 15.3%) were associated with *Mycoplasma pneumoniae* and respiratory infections.

### 2.13. Diagnostic Accuracy of Biomarkers

No studies included reported adhering to the Standards for Reporting Diagnostic accuracy studies (STARD) guidelines [63]. Within the included 36 studies, 12 (12/36; 33.3%) studies included sensitivity and specificity metrics for their biomarkers. The summary of diagnostic accuracy metrics can be found in Table 7. Articles reporting diagnostic accuracy were mostly published in 2021 (4/12; 33.3%). However, studies reporting metrics have been published between 2012 and 2025. Of the studies reporting, nine also included an area under the curve value (9/12; 75%), and six studies reported cut-off values for their biomarkers (6/12; 50%). Reporting diagnostic accuracy characteristics such as sensitivity and specificity did not equate to reporting all diagnostic accuracy metrics. None of the included studies reported positive or negative predictive values for their biomarkers. The miRNA biomarker range in sensitivity across these 12 studies was from 70% to 100%, and the specificity ranged from 72% to 100%. The range for the area under the curve across the nine studies reporting the value was 0.62 to 0.99.

**Table 7 ncrna-11-00071-t007:** miRNA biomarkers reporting diagnostic accuracy metrics.

Author [Citation]	Bacterial or Viral	Sensitivity	Specificity	AUC	Cut off Value
Jone et al. [43]	Bacterial + Viral	76.9%	76.9%	0.769	N/A
Zhang et al. [26]	Bacterial	80.2% (with PCT)	76.1% (with PCT)	0.904 (with PCT)	1.202
Mao et al. [28]	Bacterial	80.2%	85.2%	0.895	1.254
Liu et al. [30]	Viral	75%	88.04%	0.8675	N/A
Ernst et al. [47]	Bacterial	Cord plasma = 62.5% Umbilical cord tissue = 100%	Cord plasma = 90% Umbilical cord tissue = 90%	Cord plasma = 0.79 Umbilical cord tissue = 0.99	Cord plasma = −8.13 Umbilical cord tissue = −6.79
Huang et al. [31]	Bacterial	miR-497 = 95.65% FABP3 = 88.89% GPBB = 82.61% cTnI = 87.50%	miR-497 =83.33% FABP3 =94.12% GPBB = 83.33%, cTnI = 90.91%	N/A	miR-497 =2.03 FABP3 = 6.23 ng/mL GPBB = 4.01 ng/mL cTnI = 1.23 ng/mL
Qi et al. [33]	Viral	93.1%	72%	0.872	−13.03
Min et al. [60]	Viral	100%	88.89%	N/A	N/A
Cui et al. [40]	Bacterial	miR-202 = 97.4% miR-576-5p = 86.8% miR-342-5p = 77.8% miR-487b = 73% miR-206 = 48.6% miR panel = 97.4%	miR-202 = 87.2% miR-576-5p = 97.1% miR-342-5p = 97.1% miR-487b = 91.4% miR-206 = 100% miR panel = 94.3%	miR-202 = 0.981 miR-576-5p = 0.971 miR-342-5p = 0.870 miR-487b = 0.853 miR-206 = 0.664 miR panel = 0.980	miR-202 = −5.91 miR-576-5p = −10.64 miR-342-5p = −8.19 miR-487b = −7.94 miR-206 = −7.79 miR panel = 0.07
Gao et al. [42]	Viral	miR-425-3p = 79.63% miR + NEAT1 = 80.60%	miR-425-3p = 77.45% miR + NEAT1 = 87.3%	miR-425-3p = 0.832 miR + NEAT1 = 0.901	N/A
Hamdy et al. [59]	Bacterial	miR-182-5P = 62% miR-590-3p = 70%	miR-182-5P = 100% miR-590-3p = 100%	miR-182-5P = 0.620 miR-590-3p = 0.700	N/A
Qi et al. [41]	Viral	83.3%	89.8%	0.909	N/A

## 3. Discussion

The studies relating to paediatric infection included within this scoping review were conducted between 2012 and 2025, with most published between 2021 and 2023. This reflects that miRNA research remains in its infancy following its discovery in *C. elegans* in the early 1990s [64]. Due to the infancy of miRNA research, it remains a highly specialised field with minimal miRNA biomarker research for paediatric infection. Interestingly, the studies included were primarily conducted in Asia (72.2%), with the majority originating in China (55.6%). The remaining studies were conducted in the United States (22.2%) or Europe (8.3%). This may reflect the funding priorities of different regions, namely the China Precision Medicine Initiative, which was launched in 2015 [65], that may have supported miRNA biomarker discovery in paediatric infection.

The risk of serious bacterial infections varies with age, and the included articles were well distributed across all paediatric age ranges with relatively even numbers of articles relating to neonatal, infant, and toddler studies (ranging between 19.4% to 25%). There was a slight increase in publications relating to older children (30.6%). This relatively balanced distribution of ages across publications is encouraging, as each age group encounters unique physiological challenges that influence both their susceptibility to infection and potentially their miRNA response to infection. Unfortunately, none of the studies comprised cohorts that spanned all the age ranges, which removes the opportunity to assess the influences of age and infection types on alterations in miRNA biomarkers.

The studies included within this review were primarily conducted with paediatric inpatients (97%). This is likely to have proven an easy first step in miRNA discovery. The issue with this approach is that the results may not reflect the clinical setting where they are intended for use. The next phase of miRNA research needs to focus on different populations such as including emergency presentations or sequential monitoring of specific cohorts of sick children. The generalisability of the results was often confounded further as only 19.4% of the studies investigated multiple sources of infection/undifferentiated illness. The majority of the studies investigated specific confirmed infections compared to healthy controls (66.7%). The case–control designs make the discovery of differentially expressed miRNA more achievable but risk identifying non-specific immune activation markers. Future research needs to assess which miRNAs are differentially expressed compared to different infectious populations with the use of febrile controls as opposed to healthy controls.

In addition to variations in the populations and settings, the methodologies employed were diverse and non-standardised across the 36 studies. RT-qPCR investigation was used by 75% of the studies, followed by microarray for 27.7%, and next-generation sequencing for 22.2%. Techniques such as PCR and microarray are typically closed or moderately open hypothesis approaches, limiting the discovery of novel biomarkers. Fortunately, the financial barriers to next-generation sequencing are decreasing [18], which could drive more widespread adoption of this methodology in future research. Paediatric-specific protocols for next-generation sequencing are also being developed [66], which may increase the productivity of miRNA biomarker discovery for children. For all types of miRNA investigation, samples must first have their total RNA extracted. Unfortunately, there is no consensus on the optimal methodology for RNA extraction from different biofluids. The subsequent results from the studies are therefore difficult to compare as each RNA extraction kit will hold its own biases, significantly altering the miRNA profiles extracted [67,68]. The most common sample types were serum, nasal and throat swabs, saliva, and plasma, making up 77.8% of the sample types investigated. Studies investigating circulating miRNA were relatively even between serum and plasma samples (27.8% and 27.3%, respectively), and there were fewer whole blood studies (16.7%). Although plasma and serum are the preferred sample type choice for circulating miRNA discovery, serum can contain an altered miRNA profile to plasma due to the clotting process releasing miRNA from platelets [69]. However, the miRNA content of plasma can also be altered due to different processing times post-phlebotomy [69]. The consensus is that both plasma and serum are suitable for circulating miRNA studies, but results should not be directly compared [69]. Generally, whole blood is avoided in circulating miRNA biomarker discovery because its cellular material is rich in miRNA, which can interfere with the overall miRNA profiles detected [67].

A further challenge with miRNA acellular biofluid investigations is the lack of standardised normalisation miRNA for PCR techniques [70]. This contrasts with working with cellular samples, for which consistent housekeeping genes can be relied upon for normalisation [71]. A total of 54.8% of the studies reported using non-miRNA references, of which U6 accounted for 45.2%. Those included were small nucleolar RNAs (snoRNAs), RNU24, RNU43, and RNU6B (U6), which are considered poor candidates for normalisation due to disease-specific dysregulation [72], and their lack of stability in biofluids [73]. Additionally, miRNAs (18–25 nucleotides) are much smaller than snoRNAs (60–300 nucleotides), leading to differences in extraction efficiency, as methods often favour smaller RNAs. Therefore, miRNA normalisation markers should be preferred over non-miRNA molecules. The impacts of using snoRNAs as normalisation markers within these studies significantly reduces the reliability of the results and their interpretation. Spike-in RNA of a similar nucleotide length to endogenous miRNAs was used for normalisation in 16.1% of the studies. This method reduces variation more effectively than snoRNAs but does not account for differences in sample handling or processing. This may therefore introduce non-biological differences between the samples and significantly impact the data interpretation [74]. Global mean normalisation, used in 12.9% of studies, helps address issues with non-endogenous molecules by averaging endogenous miRNA across samples. However, it may over-normalise low expression samples, potentially masking true biological differences. Therefore, the best approach is to identify cohort-specific endogenous miRNA within the specific experimental samples. Within the included studies, 19.4% reported the use of endogenous miRNA; however, they did not report the assessment of the stability of their chosen normaliser within their specific cohorts.

In the context of clinical diagnostics, there is a critical need for rapid, streamlined workflows that can be readily automated to accommodate high-throughput demands. Given that approximately 75% of miRNA biomarker discovery studies to date have employed RT-qPCR, one viable route for clinical translation is the development of algorithm-based approaches. RT-qPCR can be implemented within existing clinical laboratory infrastructure, with raw Ct values subsequently normalised using either a validated endogenous reference miRNA or an exogenous spike-in control of a similar length. This normalisation enables consistent quantification across samples and facilitates the simultaneous detection of miRNA panels, with the normalised values serving as inputs to a diagnostic algorithm. The algorithm can then generate a single composite risk score, offering a clinically interpretable output to support diagnostic decision-making. Like RT-qPCR, digital PCR (dPCR) would still require a diagnostic algorithm to interpret miRNA expression levels. However, dPCR offers key advantages, including absolute quantification without the need for reference genes or standard curves, and greater precision at low input levels [75], making it especially suitable for plasma-derived miRNAs from paediatric samples. By partitioning reactions into thousands of nanolitre-sized compartments, dPCR enables direct molecule counting, reducing technical variability and improving reproducibility across laboratories [76]. It can also support multiplexed detection of miRNA panels [75], which is essential for clinical diagnostics. Adoption is currently limited by higher costs, driven by specialised equipment, proprietary reagents, and lower throughput, but as platforms become more accessible, dPCR holds strong potential for enhancing the standardisation, sensitivity, and clinical utility of miRNA-based diagnostics.

Novel miRNAs demonstrate clear potential as diagnostic biomarkers but current evidence regarding diagnostic test accuracy is lacking. Only 12 (33.3%) studies reported sensitivity and specificity data on their novel biomarkers. Within these studies, nine (83.3%) studies reported an area under the curve value and a separate six (50%) reported cut-off values for their biomarkers. None of the studies included within the review reported positive or negative predictive values. The lack of diagnostic metrics reported may unintentionally exaggerate the novel biomarkers’ clinical potential and substantially limit the understanding of the marker’s discriminatory power. These issues are reflected in that none of the 12 studies reported test accuracy in compliance with STARD guidance [63]. To improve consistency and comparability, future research must standardise approaches to population sampling, biological fluid handling, and the reporting of outcomes. All miRNA biomarker research outputs should aim to include full reporting of sensitivity, specificity, area under the curve, and positive and negative predictive values. This should also include an agreement of reference standards, and the inclusion of comparator biomarkers such as CRP and PCT. Additionally, researchers may benefit from using the Biomarker Toolkit [17], which is designed to support successful biomarker translation, and acts as a framework to improve study design and strengthen outcomes.

There were 16 miRNAs that were identified more than once across the studies, with the most commonly identified being hsa-miR-155 (6/36; 16.7%), hsa-miR-29 (5/36; 11.1%), hsa-miR-206 (/36; 11.1%), hsa-miR-142-3p (3/36; 8.3%), hsa-miR-182-5p (3/36; 8.3%), and hsa-miR-363-3p (3/36; 8.3%). There was an imbalance between bacterial- and viral-associated miRNA validations; however, this is likely to do with more studies relating directly to acute viral infection in children. There were three validated bacterial-associated miRNAs: hsa-miR-182-5p, hsa-miR-363-3p, and hsa-miR-206. There were nine validated viral-associated miRNAs: hsa-miR-155, hsa-miR-29, hsa-miR-155-5p, hsa-miR-150-5p, hsa-miR-140-3p, hsa-miR-142-3p, hsa-miR-149-3p, hsa-miR-210-3p, and hsa-miR-34a-5p. There were four miRNAs that were validated across all miRNAs identified, but not in the distinct bacterial and viral categories: hsa-miR-101-3p, hsa-miR-142-5p, hsa-miR-183-5p, and hsa-miR-486-3p. These miRNAs replicated across multiple studies, regardless of infection category, are more likely to represent non-specific infection markers. Conversely, miRNAs associated exclusively with acute bacterial or viral infections are more likely to be of clinical interest.

## 4. Materials and Methods

### 4.1. Protocol

The review adhered to the PRISMA-ScR guidelines [77] and followed the published protocol [78], which can be found at MedRxiv doi: https://doi.org/10.1101/2025.04.11.25325664.

### 4.2. Eligibility Criteria

The study included published prospective and retrospective cohort studies reporting host miRNA associated with bacterial and viral infections in English. Children aged between 0 and 18 years were included in the analysis. Studies reporting miRNAs in comparison to healthy controls, infection states compared to other infection states, or individuals of different degrees of infection were included. The inclusion and exclusion criteria are shown in Table 8.

### 4.3. Information Sources

The databases searched include Medline All and Web of Science, alongside searching for grey literature including Google, Google Scholar, and open Theses. The date the searches were completed was 19 May 2025. All screened articles were written in English. The search strategy is provided in Appendix A.

### 4.4. Selection of Sources of Evidence

The citations identified in the searches were uploaded into Rayyan [79], a screening and data extraction tool for undertaking systematic or scoping reviews. Two reviewers (OR and ADB) screened titles and abstracts, while one reviewer (OR) screened the selected articles for full-text review.

### 4.5. Data Charting Process and Data Items

Following full-text review, the included studies’ data was extracted into an Excel spreadsheet. The following items were extracted: title, year of publication, authors, DOI, date of publication, country of study, population (e.g., children with fever), setting (e.g., emergency department, UK), clinical question (e.g., bacterial vs. viral), main study methods, Next-Generation Sequencing kit, RNA extraction kit, normalisation, ages of patients, % female, total sample size, sample size study focus, sample size comparator, bacterial numbers and infections, viral numbers and infections, inclusion of CRP/PCT, reference standard, conditions of statistics, biomarkers, biomarker sensitivity %, biomarker specificity %, AUC, positive predictive value, negative predictive value, biomarker cut-off value, sample type, and sample volume.

### 4.6. Synthesis of Results

The data were synthesised descriptively, with findings presented as percentages of the total number of included studies. In some cases, with studies reporting a specific metric, the reported ranges of specific metrics were also reported. Studies were grouped according to key variables such as infection type (bacterial or viral), methodology, miRNA biomarkers, or diagnostic metric.

## 5. Conclusions

This scoping review highlights that miRNA research in paediatric infection is a novel and developing field. Children exhibit a diverse range of developing physiologies and therefore it is expected that their miRNA biomarkers may change with age. The evenly distributed age groups across the included studies suggest that no group has been disproportionately emphasised or overlooked. The research to date has focused on hospitalised children and most of the studies have focused on specific infections rather than undifferentiated febrile illness. None of the studies reported test accuracy metrics in accordance with STARD guidance and there is a lack of standardisation for how miRNA test accuracy is reported. The primary miRNA detection methodology was PCR, followed by microarray, and then next-generation sequencing. The normalisation techniques in the studies were also generally poor, with many using the small nuclear RNA U6, which is unstable in biofluid samples.

Overall, 17 miRNAs across the 36 studies included within the review were validated more than twice. Three miRNA, hsa-miR-182-5p, hsa-miR-363-3p, and hsa-miR-206, were validated to be associated with bacterial infection in children. Nine miRNAs, hsa-miR-155, hsa-miR-29a-3p, hsa-miR-155-5p, hsa-miR-150-5p, hsa-miR-140-3p, hsa-miR-142-3p, hsa-miR-149-3p, hsa-miR-210-3p, and hsa-miR-34a-5p, were validated to be associated with viral infection. A further six miRNAs, hsa-miR-101-3p, hsa-miR-142-5p, hsa-miR-182-5p, hsa-miR-183-5p, hsa-miR-363-3p, and hsa-miR-486-3p, were validated regardless of infection category but are assumed to most likely be general immune activation biomarkers. Across the 12 studies reporting diagnostic accuracy metrics, miRNA biomarkers exhibited a sensitivity ranging from 70% to 100%, and a specificity ranging from 72% to 100%. The area under the curve across the studies demonstrated a range from 0.62 to 0.99. This strongly suggests that miRNA biomarker discovery, with more rigorous study design, could have good potential for paediatric infection diagnostics in the future. However, due to the lack of adherence to the STARD guidelines and the large heterogeneity between the included studies, no clinically meaningful conclusions can be made at this time.

## 6. Research Gaps

From the synthesis of results generated within this scoping review, there are some research gaps within the field of paediatric infection diagnostics utilising miRNA targets. Firstly, there is a global incentive to increase research outputs on paediatric miRNA during acute infection. Although there is a growing interest in the field, the literature is still scarce, and the general knowledge of miRNA during infection and health of children is poorly understood. Most of the studies within this review have been conducted in Asia; there was also eight studies in the United States, and only three studies were conducted in Europe. It is possible that miRNA signatures may differ depending on the race of the child, and this should be more extensively explored to encompass all races of children to reduce bias. This review also highlights that while the included studies collectively cover all age groups relatively evenly, individual studies do not encompass the full age ranges present in paediatrics. This may limit the understanding of whether the identified miRNAs of interest are influenced by age.

With the accessibility of next-generation sequencing increasing for researchers, there should also be a focus on miRNA discovery via open hypothesis methods rather than moderately closed or closed hypothesis. This removal of pre-defined assumptions by focusing on next-generation sequencing will likely result in the discovery of more novel potential candidates with reduced bias. If next-generation sequencing is also prioritised, the identification of cohort-specific normalisation markers can also be enhanced, ensuring that the best possible normalisation can occur for biofluid samples. Following rigorous biomarker discovery, it is crucial that researchers also prioritise the validation of promising candidates. Currently, biomarker research is marked by an oversaturation of discovery efforts, with few candidates reaching clinical application, often due to flawed study designs and insufficient follow-up validation. Future research should include the development of standardised protocols tailored to specific sample types, including RNA extraction, sequencing, and PCR analysis. Such standardisation would enhance comparability across studies and increase the likelihood of successful clinical translation.

Given the global health priority of improving antimicrobial stewardship, there is a clear need for increased research into bacterial infection-specific biomarkers in children. Future studies should encompass a broad range of bacterial infections rather than focusing on individual pathogens. Such an approach would provide greater confidence that an miRNA-based point-of-care test could offer clinicians valuable guidance on antibiotic treatment decisions. Included within these improved study designs should be a more standardised methodology for reference standard choices, where there are clear definitions of infection, based on clinical and laboratory data. Finally, significant efforts should be made to include fully expansive diagnostic accuracy testing of the novel biomarkers, adhering to the STARD guidelines, and their comparisons to current testing biomarkers. Without this, there will be no progression of these novel biomarkers towards clinical use.

## Figures and Tables

**Figure 1 ncrna-11-00071-f001:**
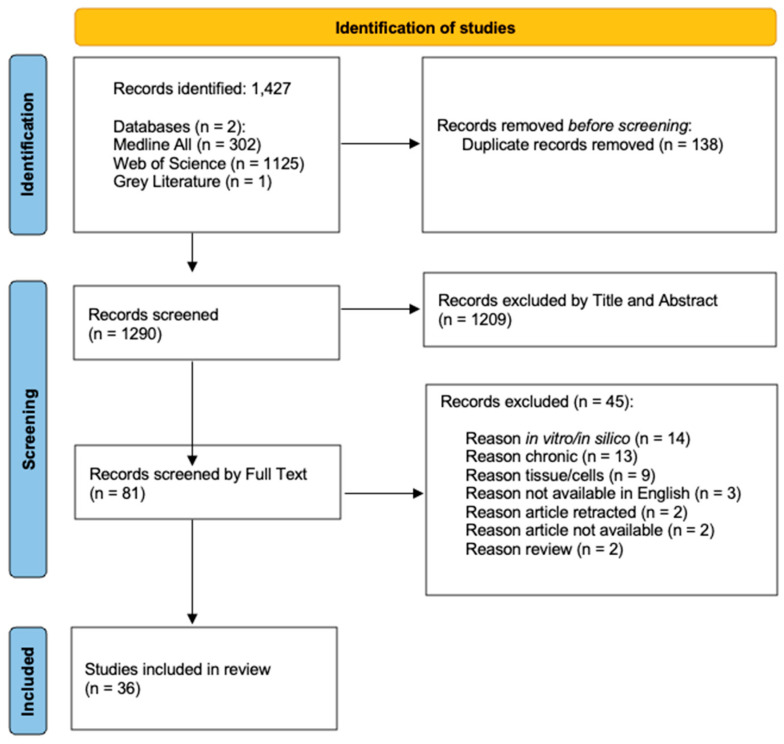
PRISMA-ScR flow diagram of screening process.

**Table 1 ncrna-11-00071-t001:** Characteristics of total included studies.

Variable		Range	Number of Studies
Age (years)		0–13	
Gender (% Female)		14–67	
Sample size	Total	6–575	
	Study focus size	3–575	
	Study comparator size	3–102	
Setting	Hospital		n = 35
	Outside of Hospital		n = 1
Clinical Question	Disease vs. Healthy		n = 24
	Disease vs. Disease		n = 9
	Severity		n = 3
Include Bacterial infections			n = 14
Include Viral infections			n = 23
Study methods	RT-qPCR methods		n = 27
	NGS		n = 8
	Microarray		n = 9
Sample type	Serum		n = 10
	Nasal/Saliva/Throat		n = 9
	Plasma		n = 9
	Whole Blood/PBMCs		n = 6

**Table 3 ncrna-11-00071-t003:** The 16 miRNAs validated across 2 or more studies, with associated diseases, number of supporting sources, and links to bacterial or viral infection.

miRNA	Conditions Associated	Number of Sources	Bacterial Associations	Viral Associations
hsa-miR-155	MRSA Pneumonia Viral Respiratory Infections RSV Infection Human Herpesvirus 6-Associated Encephalitis/Encephalopathy Epstein—Barr Virus Infection Rhinovirus Infection	6	1	5
hsa-miR-29	Severe Bronchiolitis Adenovirus Pneumonia Upper Respiratory Infections Severe Human Enterovirus 71 (EV71) Infection Mycoplasma Pneumoniae Pneumonia	5	1	4
hsa-miR-206	Early-Onset Neonatal Sepsis Human Herpesvirus 6-Associated Encephalitis/Encephalopathy Human Herpesvirus 6 (HHV-6): Acute Encephalopathy B. Pertussis	4	1	3
has-miR-142-3p	Viral Encephalitis Early-Onset Neonatal Sepsis Adenovirus Pneumonia	3	1	2
hsa-miR-182-5p	Early-Onset Neonatal Sepsis Adenovirus Pneumonia Neonatal Sepsis	3	2	1
hsa-miR-363-3p	Early-Onset Neonatal Sepsis Varicella	3	2	1
hsa-miR-155-5p	Rhinovirus Infection Epstein-Barr virus infection	2	0	2
hsa-miR-101-3p	Neonatal Sepsis Adenovirus Pneumonia	2	1	1
hsa-miR-140-3p	Upper Respiratory Infections Human Herpesvirus 6-Associated Encephalitis/Encephalopathy	2	0	2
hsa-miR-142-5p	Epstein—Barr Virus Infection Early-Onset Neonatal Sepsis	2	1	1
hsa-miR-142-3p	Viral Encephalitis Early-Onset Neonatal Sepsis	2	1	1
hsa-miR-150-5p	Early-Onset Neonatal Sepsis Severe Human Enterovirus 71 (EV71) Infection	2	1	1
hsa-miR-183-5p	Early-Onset Neonatal Sepsis Congenital Cytomegalovirus Infection	2	1	1
hsa-miR-210-3p	Kawasaki and Febrile Non-Kawasaki Children Congenital Cytomegalovirus Infection	2	0	1
hsa-miR-34	Epstein—Barr Virus Infection Upper Respiratory Infections	2	0	2
hsa-miR-486-3p	Adenovirus Pneumonia Neonatal Sepsis	2	1	1

**Table 6 ncrna-11-00071-t006:** miRNA associations with bacterial and viral infection sub-types.

Viral miRNA Associations				Bacterial miRNA Associations		
Respiratory Syncytial Virus	Human Adenovirus	Rhinovirus	Systemic Viral Infection	Sepsis		*Mycoplasma pneumoniae* + Respiratory Infections
let-7d miR-100-5p miR-10a-3p miR-125b-2-3p miR-125b-5p miR-1271-5p miR-155 miR-16 miR-203a miR-26b miR-30b-3p miR-31 miR-991-5p miR-99b-3p	miR-29b-3p let-7e-5p miR-101-3p miR-103a-3p miR-103b-5p miR-127-3p miR-142-3p miR-150-5p miR-182-5p miR-186-5p miR-27a-3p miR-29a-3p miR-342-3p miR-370-3p miR-379-5p miR-381-3p miR-409-3p miR-450a-5p miR-486-3p miR-486-5p miR-493-5p miR-494-3p miR-98-5p miR-99b-5p	miR-149-3p miR-155 miR-155-5p miR-197-3p miR-197-5p miR-21 miR-296-3p miR-504-3p	let-7b-5p let-7i-5p miR-122 miR-126-3p miR-132 miR-140-3p miR-142-3p miR-148a miR-151a miR-155 miR-183-5p miR-197 miR-206-3p miR-210-3p miR-221 miR-363 miR-381-3p miR-483-5p miR-499a-5p miR-629 miR-664a-3p miR-425-3p	1. miR-101-3p 2. miR-1184 3. miR-122 4. miR-1299 5. miR-132 6. miR-135a 7. miR-136-3p 8. miR-142-3p 9. miR-142-5p 10. miR-146b-5p 11. miR-148b-3p 12. miR-150-5p 13. miR-15a-5p 14. miR-181d-5p 15. miR-182-5p 16. miR-183-5p 17. miR-185 18. miR-193a-5p 19. miR-202-3p 20. miR-206 21. miR-208a 22. miR-211-5p 23. miR-223 24. miR-223-3p 25. miR-223-5p 26. miR-331-5p	27. miR-3613-5p 28. miR-363-3p 29. miR-375 30. miR-450a 31. miR-455-5p 32. miR-486-3p 33. miR-489 34. miR-491-3p 35. miR-497 36. miR-509-3-5p 37. miR-532-5p 38. miR-548a-3p 39. miR-549 40. miR-551b 41. miR-611 42. miR-618 43. miR-620 44. miR-629-5p 45. miR-637 46. miR-644 47. miR-720 48. miR-761 49. miR-886-3p 50. miR-590-3p	miR-1323 miR-155 miR-202 miR-206 miR-29c miR-342-5p miR-487b miR-492 miR-576-5p

**Table 8 ncrna-11-00071-t008:** Inclusion and exclusion criteria for study search strategy.

Inclusion Criteria	Exclusion Criteria
0–18 years	Parasite, fungal, or sexually transmitted infections
Bacterial or Viral infection	Cancer-related miRNA
Host miRNA	Pathogen miRNA
Biofluid samples	Animal models, in vitro, tissue samples, and in silico only studies
Human	Pregnancy-associated/foetal miRNA

## Data Availability

The data used and analysed are available from the corresponding author on reasonable request.

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
