# Peer review of "microRNA Biomarkers in Paediatric Infection Diagnostics—Bridging the Gap Between Evidence and Clinical Application: A Scoping Review"

_ncrna, 2025, doi:10.3390/ncrna11050071_

Round 1
Reviewer 1 Report
Comments and Suggestions for Authors
The manuscript by Rodgers et al. is a literature review aimed at summarizing data from the literature on the role of microRNAs in clinical diagnosis of infections. What makes this review unique is its contextualization in clinical practice and the parallel drawn with historically used markers (CRP, PCT) despite their limitations. The focus on the pediatric population is particularly interesting given the difficulties in identifying infectious causes during inflammatory processes in these patients. I have not found any other review of this type in the literature, which makes it useful to the community not only for its content but also for its methodology.
The review is well written and particularly reader-friendly despite the complexity of the subject. The rationale and importance of the subject are well highlighted.
However, I have a few comments to make in order to clarify certain points:
- Although the limitations of CRP and PCT are well explained, I feel that the lack of major advances in biomarkers over the past 30 years is not sufficiently addressed. Technological improvements should have enabled this type of advancement. An explanation of the difficulties involved in developing new biomarkers for clinical use would further highlight the importance of the study.
- Beyond the direct analytical performance of miRNAs, it would seem relevant to mention their advantage in terms of physicochemical stability within biological samples. This makes it possible to overcome certain pre-analytical constraints.
- In section 2.4, only the term PCR is mentioned. It would be useful to know whether the same type of technique is used each time or whether the studies are heterogeneous in this respect (qualitative or quantitative PCR, relative/standardized/absolute quantification, real-time or endpoint PCR, digital PCR?).
- In section 2.8, the authors state that U6 has often been used without explicitly stating this information for endogenous human miRNAs.
- In general, the results data do not specify whether the miRNAs are human or pathogen miRNAs. This may be confusing for readers who are unfamiliar with the subject. Moreover, it is surprising that few pathogen miRNAs are identified.
- Table 5 summarizes the data very well. It would be helpful to have a reverse listing with all miRNAs of interest and, for each miRNA, the pathologies in which they are altered. This would highlight the conditions that can alter the sensitivity and specificity of biomarkers.
- Table 6 (which, incidentally, has been inserted in the wrong place in the text) does not indicate whether the diagnostic performance of each study relates to bacterial or viral infections
- I regret not finding any medical-economic rationale in the discussion, which I believe is the main reason why CRP and PCT are overused despite the possibilities offered by molecular biology.
- I also think that the issue of developing digital PCR and absolute quantification techniques should be addressed when discussing the limits of standardization.
- Finally, a figure or graphic summary based on the last paragraph of the discussion would summarize and highlight the important findings of the review.
Author Response
Comment 1: Although the limitations of CRP and PCT are well explained, I feel that the lack of major advances in biomarkers over the past 30 years is not sufficiently addressed. Technological improvements should have enabled this type of advancement. An explanation of the difficulties involved in developing new biomarkers for clinical use would further highlight the importance of the study.
Response 1: Thank you for this comment. We agree and have added “The growing application of omics approaches has led to the identification of many potential biomarker candidates across a range of clinical disciplines; however, their integration into routine clinical practice has been slow. Significant hurdles have impacted the success of novel diagnostics for clinical use. These include large discrepancies in the diagnostic criteria used for clinical research and the consequential variations in biomarker discovery results and reliability. Furthermore, when promising candidates have been identified, these markers are rarely validated in large external cohorts. Even when validation studies have been conducted, a key barrier to clinical implementation is the lack of protocol consistency in validation studies. This prevents meaningful cross-study comparisons, impedes biomarker prioritisation, and ultimately stagnates the clinical biomarker outputs” to lines 76-86
Comment 2: Beyond the direct analytical performance of miRNAs, it would seem relevant to mention their advantage in terms of physicochemical stability within biological samples. This makes it possible to overcome certain pre-analytical constraints.
Response 2: Thank you for the comment, we have added “Furthermore, miRNAs demonstrate excellent physicochemical stability within biofluid samples, particularly in whole blood, serum and plasma, which makes miRNAs particularly suitable for clinical diagnostics” to lines 102-104
Comment 3: In section 2.4, only the term PCR is mentioned. It would be useful to know whether the same type of technique is used each time or whether the studies are heterogeneous in this respect (qualitative or quantitative PCR, relative/standardized/absolute quantification, real-time or endpoint PCR, digital PCR?).
Response 3: Thank you, clarification here is a good idea! We have added “Within the 36 included studies, RT-qPCR was the most common methodology (27/36; 75%). The majority of the studies utilised relative RT-qPCR (26/27; 96.3%), with the remaining study utilising absolute RT-qPCR. This was” to lines 195-196
Comment 4: In section 2.8, the authors state that U6 has often been used without explicitly stating this information for endogenous human miRNAs.
Response 4: Apologies, these were added to the table but not the text. This has now been amended to “Endogenous normalisers included: hsa-miR-320, hsa-miR-204-5p, hsa-miR-23a-3p, and hsa-miR-16(-5p) was used 3 times.” to lines 239-240
And
“including cel-miR-39 and cel-miR-238.” To lines 242-243
Comment 5: In general, the results data do not specify whether the miRNAs are human or pathogen miRNAs. This may be confusing for readers who are unfamiliar with the subject. Moreover, it is surprising that few pathogen miRNAs are identified.
Response 5: It is part of the inclusion/ exclusion criteria to only include Human miRNA (Table 8). For clarification, we have added: “human” to line 11 and “human” to line 258
Comment 6: Table 5 summarizes the data very well. It would be helpful to have a reverse listing with all miRNAs of interest and, for each miRNA, the pathologies in which they are altered. This would highlight the conditions that can alter the sensitivity and specificity of biomarkers.
Response 6: We thank the reviewer for this suggestion. Table 5 is purposefully designed to show broader associations, focusing on the most commonly reported groupings. Given the heterogeneity of the studies included in this review, we felt it was important not to overstate the certainty of individual miRNA-condition associations. Many of the miRNAs have not yet been consistently reported across multiple studies, and presenting each miRNA alongside specific conditions could risk misleading readers. A full reverse table of all 164 miRNAs would therefore be difficult to interpret and reduce readability. As a compromise, we have provided a focused table of the 16 miRNAs that have been validated across multiple studies and their associated conditions, which we believe offers a clearer and more robust overview. This can be found in the new Table 3.
Comment 7: Table 6 (which, incidentally, has been inserted in the wrong place in the text) does not indicate whether the diagnostic performance of each study relates to bacterial or viral infections
Response 7: Updated to include bacterial or viral associations
Comment 8: I regret not finding any medical-economic rationale in the discussion, which I believe is the main reason why CRP and PCT are overused despite the possibilities offered by molecular biology.
Response 8: Thank you for the suggestion we have added “A further complication arises from the high demands placed on clinical biomarkers. To be considered for implementation, candidates must demonstrate substantial improvements over existing tests in heterogeneous populations, while also offering a favourable cost–benefit profile. In the context of infection diagnostics, this means outperforming established markers such as CRP and PCT, which are inexpensive, readily available, and often considered ‘good enough' for continued clinical use. This is highlighted by the routine use of PCT in some healthcare systems, particularly in the United States. However, PCT is not routinely implemented in the UK due to a lack of cost-benefit ratio supporting widespread use.” To lines 86-94
Comment 9: I also think that the issue of developing digital PCR and absolute quantification techniques should be addressed when discussing the limits of standardization.
Response 9: This is a great suggestion, thank you! We have added “In the context of clinical diagnostics, there is a critical need for rapid, streamlined workflows that can be readily automated to accommodate high-throughput demands. Given that approximately 75% of miRNA biomarker discovery studies to date have employed RT-qPCR, one viable route for clinical translation is the development of algorithm-based approaches. RT-qPCR can be implemented within existing clinical laboratory infrastructure, with raw Ct values subsequently normalised using either a validated endogenous reference miRNA or an exogenous spike-in control of similar length. This normalisation enables consistent quantification across samples and facilitates the simultaneous detection of miRNA panels, with the normalised values serving as inputs to a diagnostic algorithm. The algorithm can then generate a single composite risk score, offering a clinically interpretable output to support diagnostic decision-making. Like RT-qPCR, digital PCR (dPCR) would still require a diagnostic algorithm to interpret miRNA expression levels. However, dPCR offers key advantages, including absolute quantification without the need for reference genes or standard curves, and greater precision at low input levels75, making it especially suitable for plasma-derived miRNAs from paediatric samples. By partitioning reactions into thousands of nanolitre-sized compartments, dPCR enables direct molecule counting, reducing technical variability and improving reproducibility across laboratories. It can also support multiplexed detection of miRNA panels, which is essential for clinical diagnostics. Adoption is currently limited by higher costs, driven by specialised equipment, proprietary reagents, and lower throughput, but as platforms become more accessible, dPCR holds strong potential for enhancing the standardisation, sensitivity, and clinical utility of miRNA-based diagnostics.” To lines 419 -440
Comment 10: Finally, a figure or graphic summary based on the last paragraph of the discussion would summarize and highlight the important findings of the review.
Response 10: Thank you for the suggestion, we have linked this into the new Table 3.
Reviewer 2 Report
Comments and Suggestions for Authors
This study provides a valuable overview of the potential of miRNA targets in diagnosing pediatric infections. The manuscript is well written and clearly organized. Please find below a few comments for the authors’ kind consideration.
- None of the studies followed STARD; please stress the need for full reporting of sensitivity, specificity, AUC, PPV, and NPV.
- Over-reliance on healthy controls may identify non-specific markers; highlight the importance of using febrile or disease controls.
- Frequent use of U6 and other unstable references undermines reliability; recommend cohort-specific endogenous miRNA normalizers.
- Variability in RNA extraction, sequencing, and sample handling limits comparability; standardization is needed.
- Many validated miRNAs may be general immune markers; please clarify which have true bacterial vs viral diagnostic relevance.
Author Response
Comment 1: None of the studies followed STARD; please stress the need for full reporting of sensitivity, specificity, AUC, PPV, and NPV.
Response 1: Thank you for this comment, we have added “All miRNA biomarker research outputs should aim to include full reporting of sensitivity, specificity, area under the curve, and positive and negative predictive values.” To lines 451- 453
Comment 2: Over-reliance on healthy controls may identify non-specific markers; highlight the importance of using febrile or disease controls.
Response 2: Thank you for this comment. We feel that this is addressed here “The next phase of miRNA research needs to focus on different populations such as including emergency presentations or sequential monitoring of specific cohorts of sick children. The generalisability of results was often confounded further as only 19.4% of the studies investigated multiple sources of infection/undifferentiated illness. The majority of studies investigated specific confirmed infections compared to healthy controls (66.7%). The case-control designs make the discovery of differentially expressed miRNA more achievable but risk identifying non-specific immune activation markers. Future research needs to assess which miRNAs are differentially expressed compared to different infectious populations with use of febrile controls as opposed to healthy controls.” – lines 361-370. Please let us know if there is anything else to add here.
Comment 3: Frequent use of U6 and other unstable references undermines reliability; recommend cohort-specific endogenous miRNA normalizers.
Response 3: Thank you for this comment, we feel that this is addressed here “Therefore, the best approach, is to identify cohort-specific endogenous miRNA within the specific experimental samples. Within the included studies 19.4% reported the use of endogenous miRNA, however, they did not report the assessment of the stability of their chosen normaliser within their specific cohorts.” – lines 414- 418
Comment 4: Variability in RNA extraction, sequencing, and sample handling limits comparability; standardization is needed.
Response 4: Thank you for this comment, we feel we have adequately discussed this throughout the manuscript, particularly within lines 371-395. Please let us know more specifically areas you wish to alter.
Comment 5: Many validated miRNAs may be general immune markers; please clarify which have true bacterial vs viral diagnostic relevance.
Response 5: Thank you for your comment, without further research it is difficult to say which miRNA have true diagnostic relevance. We have done our best to summarise validated miRNAs across multiple studies, which can be found within lines 458 - 471. However, care should be taken due to the heterogeneity of the studies.